# A Review of Biophysiological and Biochemical Indicators of Stress for Connected and Preventive Healthcare

**DOI:** 10.3390/diagnostics11030556

**Published:** 2021-03-19

**Authors:** Talha Iqbal, Adnan Elahi, Pau Redon, Patricia Vazquez, William Wijns, Atif Shahzad

**Affiliations:** 1Smart Sensor Lab, Lambe Institute of Transitional Research, School of Medicine, National University of Ireland Galway (NUIG), H91 TK33 Galway, Ireland; pau.redonlurbe@nuigalway.ie (P.R.); patricia.vazquez@nuigalway.ie (P.V.); william.wyns@nuigalway.ie (W.W.); atif.shahzad@nuigalway.ie (A.S.); 2Electrical and Electronic Engineering, College of Science and Engineering, National University of Ireland Galway (NUIG), H91 TK33 Galway, Ireland; adnan.elahi@nuigalway.ie; 3CÚRAM Center for Research in Medical Devices, H91 W2TY Galway, Ireland; 4Centre for Systems Modelling and Quantitative Biomedicine, Institute of Metabolism and Systems Research, University of Birmingham, Birmingham B15 2TT, UK

**Keywords:** stress monitoring, biomarkers, biosensors, connected health, preventive healthcare, physiological measurement, biochemical measurement

## Abstract

Stress is a known contributor to several life-threatening medical conditions and a risk factor for triggering acute cardiovascular events, as well as a root cause of several social problems. The burden of stress is increasing globally and, with that, is the interest in developing effective stress-monitoring solutions for preventive and connected health, particularly with the help of wearable sensing technologies. The recent development of miniaturized and flexible biosensors has enabled the development of connected wearable solutions to monitor stress and intervene in time to prevent the progression of stress-induced medical conditions. This paper presents a review of the literature on different physiological and chemical indicators of stress, which are commonly used for quantitative assessment of stress, and the associated sensing technologies.

## 1. Introduction

In recent years, we have seen a notable increase in anxiety, depression, pathological stress, and other stress-related diseases. Generally, stress harms the physical and mental health and wellbeing of a human [1,2,3]. Particularly, chronic stress increases the chances of cardiovascular disease [4], diabetes, stroke, and obesity [5,6]. According to the statistics from the World Health Organization, stress is associated with several medical and social problems, and these problems are seriously affecting the health and wellbeing of not only adults but also children and youngsters [7]. Therefore, a vast interest has been developed to investigate the underlying mechanisms of stress and monitoring various biophysiological and biochemical responses of the body to stress [8]. A reliable biomarker or indicator of stress could provide accurate monitoring of stress, potentially enabling the prevention of pathological conditions at early stages. In the past two decades, there has been significant development in physiological and biochemical sensing technologies. These sensors provide an excellent platform for connected health solutions and preventive care for various conditions caused by or associated with stress [9,10].

Stress can be defined as a disturbance in an individual’s homeostatic balance, with which the body attempts to cope, and this is known as the stress response [11]. Stress can be acute, i.e., immediate response to a stressor, or chronic, i.e., a state caused by a constant stress stimulus [12]. Chronic stress can lead to a stage where the body can no longer achieve homeostatic balance and the individual can no longer deal with the stressors. Activation of the stress response triggers a variety of changes in the body, caused by the stimulation of the sympathetic nervous system and the inhibition of the parasympathetic system. The stress response can vary, but it generally includes the release of stress hormones that boost the alertness of the body. As a result, there is an increase in heart rate, the blood supply to the muscles, respiratory rate, skin temperature (due to higher blood circulation), and cognitive activity, among several other responses. Stress-specific hormonal responses and other biomarkers affected by the stress response are commonly used to quantitively assess or monitor stress [3,8,11].

Stress can be assessed either subjectively via structured questionnaires (and self-reporting forms), which is also standard clinical practice, or objectively by measuring various responses of the body to stress [13]. The most commonly used tools in clinical stress assessment are based on self-reported questionnaires (for example, Cohen’s Perceived Stress Scale, PSS) or self-reported visual scales (for example, Visual Analogue Scale for Stress, VASS). Biomedical researchers are more interested in using biochemical markers for detecting stress, such as cortisol and α-amylase [14], and they trigger a stress state in the subjects under test via the Trier Social Stress Test (TSST) [15]. On the other hand, there are diverse studies available that assess stress by measuring physiological signals of the body in response to stress [16]. Details of some commonly used stress assessment tests and questionnaires are presented in Table 1. The list of all acronyms is provided in Table A1 (see Appendix A).

Most of the studies reported in the literature on stress monitoring follow a similar experimental approach, where sensors collect biophysiological data in the stress and non-stress states. First, stress is induced under a controlled environment (laboratory) [17] or in real life [18] using mental arithmetic, TSST, or Stroop test. Then, various features are extracted from the sensors’ data, and machine learning (ML) or pattern recognition is used to differentiate the stress state from non-stress (or baseline). Machine learning algorithms can be divided into two basic types. The first is supervised learning, in which the input is fed along with the classification labels to the model for prediction and classification. The second is unsupervised learning, in which no labels are given at the input, and the model is designed to group the input data on the basis of some inherent patterns or similarities. Usually, the data from the sensors are recorded on the device and then transferred to a computer or the cloud for processing and analysis. In some cases, especially in a simulated driving scenario, participants’ wearable sensors are directly connected to a computer, and real-time analysis is performed during the experiment. Various ML techniques have been used for classification, for example, support vector machine (SVM) [19], Bayesian networks (BN), artificial neural network (ANN) [20], fuzzy logic, decision tree (DT) [21], and other computer-aided diagnostics (CAD) tools [22]. Further details of these machine learning algorithms are provided in Appendix B and Table A2 (in Appendix B).

The aforementioned ML methods are benchmarked against the reference obtained via the subject’s self-reported assessment form or a psychometric questionnaire. The commonly used questionnaires are the Perceived Stress Scale (PSS), Stress Response Inventory (SRI) [23], Holmes and Rahe Stress Inventory (Life Events), and COPE inventory. These techniques obtain emotional, behavioral, and cognitive stress responses, and they are used as a ground truth. Ground truth is a reference or baseline value that helps in differentiating a stress state of the subject from a non-stress state. This is valuable in developing classification models because it makes it much easier to objectively compare two different states. The drawback of using questionnaires as ground truth is that they are designed for dedicated events and are highly subjective. Moreover, these conventional questionnaires rely on events that occurred in previous weeks; thus, they lack generalization.

There is a vast literature available that shows the association of higher heart rate with stress. This change in heart rate changes the blood flow within the body. Heart rate and heart rate variability can be monitored using an or electroencephalograph (ECG) signal, while the change in blood flow can be measured through blood volume pulse (BVP), derived from a photoplethysmography (PPG) signal [24,25]. Some studies discussed the release of sweat during stress, which changes the skin conductance measured by the electrodermal activity (EDA) measurement device [21]. Muscle tension is also related to stress and is monitored using electromyography (EMG) [26,27]. Sometimes, chronic stress can also cause a mild fever (between 99 and 100 °F), as well as anxiety and restlessness. Thus, skin temperature (ST) and accelerometer (ACC) sensors can also help in detecting stress [28,29,30].

During the stress period, the body prepares itself for a “fight-or-flight” response and, thus, catecholamines are released within the body to cope with stress. Thus, measuring plasma catecholamines can also help in the assessment of stress [31]. The role of arginine vasopressin (AVP) during the acute stress response has also been widely discussed in the literature. Copeptin is considered a stable biomarker of AVP release. Copeptin increases significantly with the increase in cortisol, prolactin, and adrenocorticotropic hormones, which are directly related to the stress response of the human body. Thus, monitoring the level of copeptin and prolactin hormones in the blood can help in detecting stress [32]. Alpha-amylase is considered to be one of the major salivary enzymes and is secreted in the saliva in response to psychological stressors [33]. Cortisol is a primary stress hormone released in the bloodstream during stress and causes an increase in the glucose level [34]. Thus, monitoring cortisol levels in the blood also helps in monitoring stress level. All the abovementioned hormones are measured using different available enzyme-linked immunosorbent assay (ELISA) kits.

Table 2 shows the bio-signals that are mostly used for stress monitoring, which include biophysical and biochemical markers. Figure 1 shows the placement of different biosensing devices used for stress monitoring.

There is a considerable body of literature available on stress monitoring using physiological or biochemical responses of the human body. However, there is no consensus on the sensitivity and specificity of these biophysiological and biochemical responses for stress identification. This sensitivity and specificity may be associated with the sensitivity of the response to stress, the sensitivity of sensors, the type of stimulators, sample size in the study, the design of the experiment, and other variables [53]. Nevertheless, the sensitivity and specificity of the measurable responses to stress are critical for long-term monitoring of stress in the context of preventive and personalized care. This paper presents an up-to-date review of the literature on biophysiochemical indicators of stress with a focus on connected and preventive healthcare. In this paper, we provide summaries of the available literature on stress indicators and a critical review addressing the sensitivity and specificity of the sensors, as well as indicators of stress, in the discussion section. The rest of the article is organized as follows: Section 2 describes the search terms and inclusion criteria; Section 3 provides a brief review of the existing literature on different indicators of stress up to August 2019. Lastly, discussion and conclusions are presented in Section 4.

## 2. Search Methodology

The literature search was carried out on Google Scholar, Pub Med, and the IEEE digital library. The search terms were formed combining three general keywords (stress monitoring, stress biomarkers, and stress indicators) with a maximum of two specific keywords (sensors, signals, physiological sensors, and biochemical sensors). Initially, a total of 5081 articles were retrieved after adding up various combinations of the abovementioned search terms, and 72 papers were selected for review reporting human studies for stress monitoring using physiological or biochemical markers.

The selection of literature included in this review was divided into two stages:In the first stage, studies related to mental and physical stress monitoring using biophysiological and biochemical parameters were included by reviewing the title and abstract of the papers, which resulted in 72 articles reporting the studies on human stress monitoring using biophysiological and biochemical methods.In the second stage, out of 72 articles, 38 original studies were selected for further review. A detailed review of the 38 selected manuscripts was performed to obtain information on the type of sensors, methods of stress induction, number of subjects volunteered for experimentation, and findings of authors about the use of the specified indicators of stress. Two additional papers identified during the review stage were included in this paper for a literature review.

## 3. Review of the Papers

Figure 2 presents a list of biophysiological and biochemical indicators of stress.

### 3.1. Biophysiological Indicators

#### 3.1.1. Stress Monitoring with a Single Sensor

In the selected literature, we found only four papers that used a single biophysiological sensor to detect and monitor human stress. The highest accuracy reported was 85.3% [38], while the lowest was 80.3% [54], using a galvanic skin response (GSR) and EDA sensor, respectively. Readers are directed to the discussion section and Appendix B for more information on accuracy assessment.

Further details of studies reporting the use of a single sensor for stress monitoring are presented below.

Kim et al. [38] developed a model that could accurately classify driving stress while driving in a real environment using physiological responses of the drivers measured through GSR devices. For this purpose, the authors used two types of data. The first dataset was a driving stress dataset available from PHYSIONET that contained multiple parameter data of 17 drives by nine drivers. In this study, the authors used GSR data for only 10 out of 17 drives as others did not have clear annotation, and they developed the proposed model. Secondary data were collected through experimentation and used for the validation of the developed model. GSR data were collected through the wrist-worn device of only one driver. Driving on a highway or highway under construction was labeled as high stress, while driving in rural areas was labeled as a low-stress time-period. For the classification, the authors designed a binary logistic regression model and achieved an overall accuracy of 85.3% using data from PYSIONET, while they achieved 83.2% using the validation data, analyzed through cross-validation. The authors also proposed that their developed model can be embedded in existing wearable GSR sensor devices, thus enabling detection and monitoring of driving stress in real time.

Bitkina et al. [54] monitored the stress of one Korean driver, driving in a real environment for 60 min each day for a total of 21 days. The authors collected EDA signals and quantified the relationship between driving stress versus road type and traffic conditions. In the study, the driving route was divided into five segments, i.e., city 1, highway 1, highway 2, highway 3, and city 2 with two toll stops. While driving, the EDA signal was recorded through the Empatica E4 sensor. The authors found that the driver felt more stressed driving through the city areas and was less stressed driving on the highways. Similarly, the high volume of traffic on the road also induced driving stress. Logistic regression was used as a classification method to classify stress from the non-stress condition. From the experiment, the authors concluded that their classification results indicate that road type and traffic conditions are important features related to driving stress. They reported an accuracy of 80.3% with a sensitivity of 85%, specificity of 78%, and positive predictivity of 70%.

Li et al. [55] monitored the daily life stress of only one person. For this study, the authors monitored bathtub ECG for 6 months. During the study, ECG was measured, and analysis was done using heart rate variability (HRV). To quantify stress, a stress index was proposed. The room temperature was controlled around 20 °C, while the temperature of the water was set to 39 °C. The working diary of the subject was taken as ground truth to evaluate stress levels. By combining time- and frequency-domain nonlinear features, authors claimed to have more information that helped in estimating stress level. The variation of stress index showed high concordance with the work schedule of the subject and, thus, could provide an acceptable solution for the comparison of stress levels of different individuals. They also claimed that the stress index was significantly higher during both mental and physical stress.

Liu et al. [56] determined the feasibility of a signal parameter and developed a stress-monitoring device. The author used only EDA signals for the detection of stress. To determine the feasibility of EDA to discriminate among high, medium, and low levels of stress, the authors used a predefined database from the MIT Media Lab. They chose 11 foot-based EDA signals of 11 drivers and extracted 18 different features. Data of drivers were collected while driving on the highway, in the city, and at rest. They argued that, while their overall accuracy was less than some multisignal systems, it could be seen as a better balance between computation load and recognition performance. The authors suggested that their results provide a promising line of research to develop a practical personalized stress monitor. After using Fisher projection and a linear discriminant analysis (LDA) on the data collected from the dataset, the authors claimed to achieve a classification accuracy of 81.82%.

#### 3.1.2. Stress Monitoring with Two Sensors

In the selected literature, we found only three papers that used a combination of two sensors to detect and monitor human stress. The highest accuracy reported was 94% using multiple features with EDA and PPG sensors [19], while the lowest reported accuracy was 78.98% using EDA and respiratory rate (chest band) sensors [44]. Further details of studies reporting the use of a combination of two sensors are presented below.

Han et al. [19] proposed a detection technique that detects three levels of stress (no stress, moderate perceived stress, and high perceived stress). For the aforementioned purpose, data of ECG and respiration rate were collected. The authors collected data from 39 subjects, in total. As a stress stimulus, the Montreal Imaging Stress Task (MIST) was used. These tasks included mental and psychosocial stress factors related to any workplace. Along with SVM, they investigated linear discriminant analysis (LDA), k-nearest neighbors (kNN), and AdaBoost. The authors reported an accuracy of 84% using random forest (RF) features and an SVM classifier in discriminating three stages of stress. However, for binary classification, i.e., rest and stress, they achieved an accuracy of 94%. The authors also concluded that LDA, nearest neighbors, and AdaBoost did not perform well in terms of classification accuracy, as compared to SVM.

Sandulescu et al. [44] continuously monitored the stress of students in a real-time environment. The authors proposed a stress detection technique using data from EDA and PPG sensors. They selected five people to participate in their studies. To induce stress, the subjects performed the Trier Scope Stress Test (TSST) with a public speaking task, a cognition task, and finally a neural task. The neural task consisted of answering a predefined questionnaire with an allocated time of 2 min. The public speaking task was a 5 min job interview. After the interview, as a cognitive task, participants were asked to count backward from 1022 with a gap of 13 and, if they made a mistake, they had to restart the count. The time allocated for this task was 5 min. The goal of this experiment was to see the performance of the proposed approach as a personalized stress detection device for each subject. The authors were successful to detect the stress of each participant with an average accuracy of 78.98% across five participants using SVM. They claimed that their approach is a starting point for the detection of a subject’s stress state in real time, as well as for the treatment of people, which will eventually improve their life quality.

Mohino-Herranz et al. [57] assessed the mental fitness of different subjects participating in the study. The authors used ECG and thoracic electrical bioimpedance (TEB) signals to monitor stress. In total, 40 subjects were recruited for this experiment. Their stress induction protocol involved three stages. Low-pass filtering and decimated intermediate frequency-based algorithms, along with a multilayer perceptron classifier (MLP), were used for feature extraction and classification purposes. In stage 1, each subject was twice shown the film “Earth”, which is a BBC documentary. The second stage was about a game based on addition. A sum of two digits was displayed on the screen and the subject had to do calculation mentally and choose the correct option from the given multiple solutions. In the final stage, the subject chose one of three films i.e., “American History X”, ’Life Is Beautiful”, or “I Am Legend” to validate sadness. The proposed system was analyzed in three different ways, i.e., activity identification, emotional state, and mental activity. The parameter used for the analysis was the error rate. With MLP classifier, the authors achieved an error rate of 21.23%, 4.77%, and 32.33% for activity identification, emotional state, and mental activity, respectively. The measurement showed high potential in the use of ECG and TEB signals for the detection of long periods of stress or a sudden increment in mental work overload or emotional responses of the people, especially in those who are associated with risk, for example, police, soldier, and firefighter.

#### 3.1.3. Stress Monitoring with Multiple Sensors

In the selected literature, we found 16 papers that used a combination of more than two sensors to detect and monitor human stress. The highest accuracy reported was 97.4% using EMG, EDA, and respiratory rate sensors along with an artificial neural network as a classifier [40]. The lowest reported accuracy was 61.8% using ECG, PPG, EDA, and temperature sensors with an SVM classifier [35]. Further details are presented below.

Gjoreski et al. [21] developed a stress detection method that could measure daily life stress accurately, unobtrusively, and continuously. They proposed a context-based stress detection technique using EDA, skin temperature, PPG, and accelerometer data. The study involved five subjects. To induce stress, a series of randomly generated equations were provided to participants who were asked to solve them verbally. The time allocated to solve each equation changed dynamically, i.e., after every two consecutive correct answers, allocated time was decreased by 10%, while, for every two incorrect answers, the time was increased by 10%. The experiment protocol included three sessions of different difficulty levels, namely, easy, medium, and hard equations. As participants were solving equations, they were also asked to fill out the STAI questionnaire to measure the anxiety levels before and after each session. User activities were recorded using an accelerometer and saved as context information. The experiment showed that, without contextual information, the stress detection was not in the range of acceptable accuracy. When contextual information was included, the SVM classifier F-score jumped to 0.9 from 0.47 and precision increased to 95% from 7%. Therefore, the authors concluded that contextual information is required to distinguish real-life stress from other situations (for example, hot weather, eating, or exercising), which may induce similar physiological arousal.

Kim et al. [35] developed an emotion-recognition system using different physiological sensors. Physiological sensors such as ECG, PPG, EDA, and skin temperature were used to measure the emotions of children. For the experiment, children of ages 5 to 8 years old were recruited and were randomly divided into two groups. The first group consisted of 125 subjects, while the second had 50 subjects. For induction of stress, a multimodal approach was used which was based on some audio, visual, and cognitive stimuli, developed under the supervision of consultants from the field of physiological psychology. The authors of this article identified different factors such as physical activity, physical status, and cognitive workload that could affect the signals. The overall analysis consisted of signal preprocessing and the highly relevant stages of feature extraction and classification. For ground truth, a self-reported questionnaire was used. Subjects had to score on a scale of 1 to 5 about how they felt during the experiment. The authors reported that they achieved a recognition rate of 78.4% for three emotional and 61.8% for four emotional states using SVM as a pattern classifier.

Sun et al. [36] determined the mental and physical stress of a subject during different physical activities. The authors used ECG, GSR, and accelerometer data to determine the mental and physical stress of a subject. A total of 20 subjects participated in this experiment. Sitting, standing, and walking were the three activities used to induce physical stress. In parallel, a computer-aided Stroop color test and mental arithmetic problems based on the Montreal Imaging Stress Task (MIST) were used as mental stressors. All the tests were conducted under a time constraint and in all three activity positions, i.e., standing, walking, and sitting. After analysis, the authors showed that the inclusion of accelerometer data improved the mental detection process in a mobile environment. The decision tree classifier, 10-fold validation, and least complex classifier [58] performed best for their experimentation. The authors reported a classification accuracy of 92.4% when using accelerometer data along with other physiological signals, while 80.9% accuracy was obtained for inter-subject classification.

Chen et al. [37] developed a stress detection system for drivers based on multimodal features and kernel-based classifiers. They recorded ECG signals, GSR signals, and respiratory rate. The authors collected data of 14 drivers, driving on the prescribed route (in a real driving environment). During the experiment, physiological and behavioral (video) information was collected to determine the stress index. Moreover, a questionnaire was also used to assess the accuracy of the measurements. The authors drew three conclusions from this study. Firstly, different driving stress levels can be characterized using this specific set of physiological sensors. Second, the multimodal feature set gives more reliable and accurate drive status. Thirdly, with efficient feature selection techniques and kernel-based classifiers, a promising real-time monitoring and alerting setup can be developed and placed in cars to prevent accidents caused by the negative status of drivers. The classification was done using different kernel-based SVM models. Authors analyzed the data in terms of precision, i.e., (true positive)/((true positive + false positive)), sensitivity, and specificity. While using a full feature set, SVM with a linear kernel gave the highest inter-drive classification precision, i.e., 0.999. For cross-drive level, SVM with a radial basis function (RBF) kernel gave a precision score of 89.7%.

Healey et al. [39] analyzed four physiological indicators measured using wearable sensors, i.e., cardiac activity from ECG, muscle activity from EMG, respiratory rate from chest cavity expansion, and skin conductance from EDA to monitor the stress of in-vehicle drivers. The authors collected data from 10 drivers. A driving route was simulated, with several driving-related stress situations. The route designed for the drivers included garage exit, city road, toll booth, highway driving, ramp turnaround, two-lane merging, bridge crossing, and entering the garage. These routes were assigned stress rating from 1 to 7 on the basis of median scores of questionnaires, and stress events were labeled very high, high, neutral, or low. Data from the sensors were analyzed by extracting various signal features, including means and variances. A k-nearest neighbors (kNN) classifier was used to test an individual and a combination of multiple features for stress classification. The accuracy of predicting the stress state for each feature or combination of features was set as the test criteria. The authors found the mean respiratory rate to be the best single predictor of stress with a prediction accuracy of 62.2%. Accuracy was defined as (TP + TN)/(all observations), where TP is true positive and TN is true negative. The combination of all features from the four type of sensors had an overall accuracy of 86.6%.

Similarly, Healey et al. [40] also investigated stress incurred by drivers during on-road driving. The authors monitored four different physiological signals for analysis, i.e., ECG for heart rate, EMG for muscle activities, EDA for skin conductance levels, and respiration rate. Physiological data of 24 drivers, who drove the car for at least 50 min, were collected by the authors. After performing preprocessing and excluding redundant data, only 16 drivers’ data were selected for analysis. The driving protocol included rest, highway drive, and city drive, which was supposed to induce a low, medium, and high level of stress in the drivers. For analysis, a questionnaire and videotape recording were used. The analysis of the questionnaire was done using two rating scales. First, on a rating scale, the drivers rated the driving events (environments) from 1 (non-stressful) to 5 (stressful). On the second scale, they had to rate their feelings on a scale of 1 (least stressful) to 7 (highly stressful). Videos were also analyzed using scores based on action. Potential stressors included turns, stops, bumps, and head-turning (gaze change). The classification was done using an artificial neural network. The experiment showed that measuring signals from physiological sensors is a practicable method to quantify the drivers’ stress. They achieved a classification accuracy of 97.4% using data intervals of 5 min and found the highest correlation between heart rate and skin conductance metrics.

Shi et al. [42] conducted studies on nonintrusive, continuous stress detection. The authors used sensors to obtain ECG, GSR, respiration rate, and skin temperature as physiological signals. A total of 22 subjects were involved in this study. Each subject went under four stressors and six rest periods. Stressors periods included one public speaking test, two mathematics tests, and one cold pressure stress test. These stressors represented mental, social, and physical challenges incurred by a person during physical or emotional stress. For annotation, the authors gathered Ecological Momentary Assessment interviews before and after each stressor and rest period. After the experiment, the authors concluded that their SVM-based model detects stress with high precision and recall rate (68% accuracy), especially when they used personalized information with SVM.

Wijsman et al. [43] detected the mental stress of participants using physiological signals. They recorded respiration rate, ECG, EMG, and skin conductance signals. After pre-processing, high-quality data of 18 participants (from a total of 30 participants) were used for further analysis. Participants first filled a Perceived Stress Scale (PSS) questionnaire and then were exposed to three different stress situations, i.e., some calculation task, a logical puzzle, and a memory test. All these tests were restricted in time and ran along with some distracting news heard through headsets. For ground truth, a questionnaire was filled in before and after each test. Authors claimed to achieve accuracy of 80% for two-class classification, i.e., rest vs. stress, using the principal component analysis (PCA) technique. They concluded that such accuracy indicates the suitability of these features for the detection of stress in a subject.

Choi et al. in [45] proposed a wearable device to measure stress, drowsiness, and fatigue of the drivers. The authors measured the GSR, accelerometer, temperature, and PPG signals of 28 drivers. For stress induction, a simulator (software), i.e., City Car Driving, was used. This software allowed authors to control different driving environments such as dense traffic and accidents. The experiment was divided into four parts, normal (driving on city road), stressed (crowded road, noise, and accidents), fatigued, and drowsy (driving on the highway for 2 h with no cars around) states. All the signals were measured using one wrist-based wearable device. The authors reported an accuracy of 68.31% for the classification of four states (i.e., normal, stressed, drowsiness, and fatigue) and accuracy of 84.46% for the classification of three states (i.e., normal, stressed, drowsiness, or fatigue).

Choi et al. [59] estimated the autonomic nervous system state through an analysis of heart rate. The authors used an accelerometer (ACC), a global positioning system (GPS) unit, ECG, EDA, and respiration rate along with time recording for the experiment. Only three subjects participated in this study. For mental stress induction, subjects were asked to perform a mental arithmetic test [60] or Stroop color word test [61], while, for relaxation, each subject took deep breaths to relax. The authors collected four tests for each subject on five different days, collecting 20 test results per subject. The principle dynamic modes (PDM) method was implemented to predict the activation levels of sympathetic and parasympathetic autonomic responses. For comparison of the results obtained by PDM, power spectrum density (PSD) was calculated using Welch’s method. Authors found that the PDM algorithm shows lower inter-subject variance and interestingly shows some comparable performance between and within subjects, whereas PSD performance decreases when used between subjects.

Hosseini et al. [62] introduced a new labeling protocol of EEG signals for states of emotional stress using a picture induction model. They carried out a quantitative and qualitative analysis of PPG, skin conductance, and respiratory rate to efficiently segment EEG signal and, thus, improve the overall stress recognition system. For the experiment, 15 male student volunteers were recruited. As a pretest, a State–Trait Anxiety Inventory (STAI [63]) was conducted to determine the best psychological input. For stress induction, a subset of pictures from the International Affective Picture System (IAPS) was selected. IAPS has 956 images that evoke emotions. The authors used wavelet coefficients and Higuchi’s algorithm, as well as the correlation dimension, to extract characteristics of the EEG signal. They achieved an accuracy of 82.7% using the Elman classifier and concluded that the application of nonlinear time-series analysis on EEG signals provided insights into the dynamic nature and variability of brain signals. Results of the experiment endorsed the hypothesis that stress-monitoring systems developed with the fusion of physiological signals and EEG signals perform better in comparison to systems having these signals separately. Moreover, EEG signals showed a better classification response in time than psychophysiological signals.

Karthikeyan et al. [64] introduced a method that could detect stress using short-term signals. In this experiment, the authors used <5 min electrocardiogram and heart rate variability (HRV) signals for the detection of stress. For the experiment, 60 subjects were recruited, who participated in a Stroop word–color test with different time constraints. The designed protocol gradually increased the induced stress from relaxed to high levels. Before and after a stressful task, subjects were dispensed with 3 min of relaxation time. ECG and HRV signals were recorded for all four states incurred during the experiment. Optimum features of ECG were extracted through fast Fourier transform (FFT). Accuracies of 91.66% and 94.66% were achieved using a probabilistic neural network (PNN) and k-nearest neighbor (kNN) classifier, respectively. The result showed an improvement in the overall detection of stress in the subject independent method as compared to subject dependent studies. The authors also proposed the hypothesis that for effective assessment, the HRV signal can be useful if collected for a longer period than 25 s.

Muaremi et al. [65] investigated night sleep patterns and identified the most vital sleep parameters that could be used for stress detection. They recorded posture data with a chest-strap sensor and accelerometers on the body and arms. The study also included ECG, HRV, body temperature, respiration rate, and GSR data for analysis of biophysiological parameters. The 10 subjects recruited for this experiment were performing the Hajj pilgrimage and wore two devices, a wristband and chest strap, during the experiment. The authors interviewed the participants on the first day and collected information about their health status, making sure that they understood the experiment completely. Before going to sleep, each participant was asked to fill out a stress questionnaire. At bedtime, they were asked to turn the sensors on and to turn them off when getting up. The authors believed that, by incorporating stress estimated during the daytime, they could get a step closer to finding a complete solution for stress detection in daily life. The authors also concluded that, during sleep, the activity of the upper body and the duration of sleep (among the physical features studied) are strong indicators of stress, while ECG and HRV were the most relevant in the group of biophysiological parameters. Body temperature could be neglected as it did not contribute to any relevant conclusion.

Mozos et al. [66] presented an approach to stress detection in people who suffer from stress in social situations. Two types of sensors were used to develop social and physical responses. The sensors included EDA, PPG for calculating HRV, and a sociometric badge to record the voice of the subject. A total of 18 subjects was recruited, and a protocol based on TSST, as defined above, was designed to induce stress within each subject. In addition to TSST, participants were also asked to complete the STAI (State–Trait Anxiety Inventory) to determine their anxiety levels. The authors claimed that the sociometric device alone provided better-predicted outcomes when used together with other parameters. The reason for this could be that, in TSST, subjects continuously speak and are free to move around during the experiment. The classification was done using AdaBoost, radial basis function (RBF) kernel SVM, linear kernel SVM, and K-nearest neighbors (kNN). The authors also found that a higher accuracy, i.e., 92%, could be achieved with the SVM (RBF kernel) classifier, as compared to linear kernel SVM (80%), AdaBoost (67%), and kNN (62%), when using selected set of features.

Lee et al. [67] developed a wearable glove that could detect the stress of drivers. They recorded the PPG and inertial motion of the drivers along with the driver’s behavior. A total of 28 drivers participated in this study. For the driver’s safety, the experiment was performed on a driving simulator in the laboratory. The whole interface was synchronized with Euro truck driving simulator version 2 [68] operating on the computer. The driving scenarios were divided into three groups, the city, urban areas, and highway. The authors’ test results indicated that the sequential feature selection and an SVM classifier with RBF kernel were able to achieve a classification accuracy of 95%, which shows the suitability of their glove as the driver’s stress detection device.

Can et al. [69] developed a stress detection technique through unobtrusive wearable devices. The authors did their experimentation using PPG, EDA, GSR, and accelerometer signals. They collected data from 21 participants. To evaluate their technique, the data were collected via the INZVA algorithmic programming summer camp, organized in Istanbul, Turkey. Experiment sessions were divided into three parts i.e., training, the contest, and a free day. Context information, such as whether participants were in a lecture or contest or were enjoying free time, was known and used as a ground truth. The authors experimented with different preprocessing methods and set different hyperparameters to analyze the data. They inferred that choice of preprocessing and parameter selection depended upon the selected machine learning technique. Person-specific models gave better classification accuracy than general models. The authors suggested that, if the individual data of each person are enough to design a person-based model, it should be developed; otherwise, each person should be clustered by their behavior during stress, and then a clustering model should be developed to increase the classification accuracy of the general model.

In summary, stress response varies from person to person and cannot be detected or monitored easily. One reason could be that there are no standard ground-truth values of variables/signals to be classified as stressed values/signals. In the literature reviewed above, most of the studies reported using different questionnaires and self-reports as ground truth for stress classification. Few studies related the increased heart rate and abrupt changes in the skin conductance, respiratory rate, and blood pressure with stress and used them as ground truth. It can be observed that some studies measured the same signal(s) and used the same stressor(s) and same ground truth (method), but the reported results showed significant differences. Figure 3 shows the reported prediction accuracies versus the prediction algorithm using different stress indicators/markers. For prediction, most of the authors used SVM (with different kernels) as a predictor, while the highest prediction accuracy was achieved using an artificial neural network predictor. Table 3 summarizes the above-discussed papers and their conclusions.

### 3.2. Biochemical Indicators

#### 3.2.1. Stress Monitoring Using a Single Sensor

In the selected literature, we found five papers that used a single biochemical sensor to detect and monitor human stress. Further details are presented below.

Ullmann et al. [48] determined the relationship between chronic physical and mental stress, as well as chronic activation of the hypothalamic–pituitary–adrenal (HPA) axis. The authors collected hair samples and calculated hair steroid levels to measure stress. The authors performed a pilot study on 40 healthy participants. Hair steroid hormone level was recorded to determine anxiety, depression, physical activity, stress perception, mental burden, sense of coherence, resilience, and physical complaints. For objective stress markers, the Perceived Stress Questionnaire (PSQ) was used as a positive validation control, and results were compared with other conventional measuring techniques [70,71,72]. Regarding mental stress, the authors demonstrated that subjects reporting stress by mental burden exhibit higher and long-term concentrated levels of cortisone, dehydroepiandrosterone (DHEA), and cortisol in hair. The results reported a significant correlation of cortisone, cortisol, and DHEA with physical (*p* = 0.001) and mental (*p* = 0.034) stress and with subjectively perceived stress (*p* = 0.006). The authors concluded that the concentrations of steroids in the hair are a decisive predictor of a long-term activity increase in the HPA axis. Additionally, this biomarker can capture stress even after burdening events or any physical activity is finished.

Rohleder et al. [51] proposed the measurement of salivary alpha-amylase instead of catecholamines from blood plasma. This study aimed to determine a noninvasive (biochemical) stress indicator, other than salivary cortisol, that can be stored at room temperature and analyzed later. The authors collected saliva samples and extracted alpha-amylase and cortisol levels to detect and monitor stress. They performed two types of studies. In the first study, a total of 12 healthy subjects were investigated, and this study focused on an increase in salivary alpha-amylase. The second study was about seeking any circadian rhythm associated with salivary alpha-amylase and 17 healthy subjects were investigated. A psychosocial stress test, Trier social test (TSST), and reading test were conducted to induce stress. These tests were conducted right after, after 30 min, and after 1 h of awakening of the participant, as well as at 11 a.m., 3 p.m., and 8 p.m. Authors found that the level of salivary alpha-amylase was significantly lower in smoking females than non-smokers, while it was higher in smoking males than in non-smokers. They also identified that the production of salivary cortisol affects the association of norepinephrine and amylases. Furthermore, during stress, activation of parasympathetic nervous systems decreases the overall saliva production and volume. Therefore, the increase in induced stress may be confused with a decrease in saliva volume. Following these observations, the authors suggested that the volume of saliva and amylase levels should be measured relative to the saliva produced.

Tasaka et al. [52] investigated the relationship between chewing rate and salivary hormones produced due to stress. The authors analyzed salivary cortisol levels and EMG along with chewing rate for this experiment. Sixteen male subjects were selected for the experiment. All were staff or students at Tokyo Dental College. Subjects were first asked to rest for 30 min. For stress induction, subjects were given arithmetic problems to solve within 30 min. Salivary samples were collected immediately after the test, referred to as S1. Then subjects were asked to chew a gum, which was tasteless, for 10 min, and second samples of the saliva were collected, referred to as S2. Subjects were then allowed to rest for 10 min, and the third sample of saliva was collected, referred to as S3. Change in the salivary cortisol levels was analyzed for S1 and S2 and for S1 and S3, for slow, habitual, and fast chewing rate. The authors suggested that fast chewing had a greater effect on stress release than slow chewing rate, while the integration of EMG signals did not show any major difference in the three chewing rates.

Russell et al. [73] analyzed hair cortisol as a complementary mean of stress monitoring. The authors collected hair samples from the participants to measure stress. Cortisol levels in the urine and saliva capture stress levels in real time. However, analysis of hair cortisol can be presented as complementary means of stress monitoring. Therefore, the authors reviewed development, limitations, and some unanswered questions about the hair cortisol biomarker for chronic stress monitoring. Hair grows 1 cm per month; thus, a sample was carefully sectioned into different length segments (mostly 3 cm of hair to see the last 3 months cortisol production). To extract hair cortisol, it is finely minced with the help of scissors and incubated in a solution of methanol. This solution is then dried and reconstituted in another solvent such as phosphate-buffered saline [74]. After the extraction of the cortisol, ELISA, liquid chromatography–mass spectrometry (LC–MS/MS) or radioimmunoassay (RIA) can be used to quantify hair cortisol [75]. Immunoassays are mostly used to measure cortisol in blood, saliva, urine, and hair. These methods are change-sensitive and subject to inter-assay variability. The authors identified some gaps in the currently available literature: firstly, the understanding of the mechanism via which the cortisol is incorporated into the hair, and, secondly, the determining factors that cause variation in the hair cortisol, such as the effect of hair washing. Authors also suggested that more study is required to determine if and to what extent cortisol in hair originates from the eccrine gland (also known as merocrine glands), blood, or any sebaceous source.

Sunthad et al. [76] wanted to improve class management using a technique that enabled teachers to monitor and analyze student’s stress levels of the brain. A sensor with two channels of near-infrared spectroscopy (NIRS) was used to monitor the brain activity of four students, who participated in this experiment. Four students were recruited for the experiment, and they were asked to fill in questionnaires before and after the experiment. An NIRS HOT-1000 device was used in this experiment, placing it on the forehead of the participants to measure brain activity during a classroom scenario. This situation simulated decision-making and memory recall tests to increase stress levels. Some survey questions were also asked during the experimentation and were used as a ground truth. Mental stress levels were then divided into three categories, i.e., relaxed, medium stress, and high stress. The results showed that, whenever high stress occurred, the average value of oxy-haemoglobin (oxy-Hb) increased. Stress is directly related to emotion function. As the right hemisphere is related to emotions, while the left hemisphere plays role in logical thinking and reasoning [77], the average value of the right hemisphere was higher during high stress. The left hemisphere of the student was not significantly affected by stress tasks.

#### 3.2.2. Stress Monitoring Using Two Sensors

In the selected literature, we found three papers that used a combination of two sensors to detect and monitor human stress. Further details are presented below.

Russell et al. [78] performed an analysis of hair cortisol for stress monitoring of athletes. The authors collected samples of sweat and saliva along with hair from athletes. They collected samples of 17 subjects exposed to intense exercise and suggested that cortisol can be incorporated into hair through serum, sweat, and sebum sources. For analysis of samples from sweat and saliva, a salivary enzyme-linked immunosorbent assay (ELISA) was used. For hair, an in vitro test was conducted by exposing hair to hydrocortisone. The content of cortisol in hair was also determined by using the ELISA method. The authors concluded that human sweat contains significant cortisol levels that can be compared with salivary cortisol levels. The study pointed out that intense exercise may increase the concentration of cortisol in hair, which is not decreased by hair washing. Thus, this point should carefully be taken into account when dealing with hair cortisol as a chronic stress biomarker.

Bonke et al. [79] investigated the effect of early life stress (ELS) on autonomic (heart rate) and endocrine (salivary cortisol) indicators. The authors measured the heart rate using a PPG sensor, while functional magnetic resonance imaging (fMRI) scans were done in the author’s lab and salivary cortisol level was measured using a commercially available device named Salivette^®^ (by Sarstedt Inc., Rommelsdorf, Germany). There were 32 healthy participants in this study who were exposed to stressors, from mild to moderate. Participants were asked to perform some arithmetic tasks. After that, the process of fMRI scanning and saliva collection was explained. Two saliva samples were collected before the Montreal Imaging Stress Test (MIST), while one sample was collected after the test. The authors used the Childhood Trauma Questionnaire (CTQ) [80] for ELS. The CTQ consists of 28 different items with five self-reported ELS subscales, i.e., emotional abuse, emotional neglect, physical abuse, physical neglect, and sexual abuse. The result showed no significant effect of early life stress on heart rate (autonomic indicator) and salivary cortisol (endocrine indicator), but the authors suggested that heart rate is a better indicator then salivary cortisol, as it is more sensitive than salivary cortisol in differentiating between control and stress conditions.

Wu et al. [81] proposed a model for stress assessment using psychological stress response inventory, HRV measurements, and salivary cortisol levels. The authors used ECG signals to calculate HRV, while cortisol levels were monitored in saliva. For this experiment, data were collected from 30 college students who went through their final examination. The authors used a simplified version of the original Severe Respiratory Insufficiency Questionnaire (SRI-Q) [82], composed of 22 questions and seven stress factors. Each question was scored in five-point formats, i.e., not at all, somewhat, moderately, very much, and absolutely. Cortisol levels were determined using solid-phase radioimmunoassay (RIA), gamma counter, and Coat-A-Count cortisol kit (from Siemens, USA). The ECG signals were recorded to obtain HRV using a wearable system. The authors concluded that salivary cortisol levels are negatively correlated with an HRV indicator of parasympathetic activity, while they are positively related to an HRV indicator of sympathetic activity. The results also showed that the value of mental stress index (MSI) was very sensitive to acute stress and could predict it with an accuracy of 97%.

#### 3.2.3. Stress Monitoring Using Multiple Sensors

In the selected literature, we found four papers that used a combination of three or more sensors to detect and monitor human stress. Further details are presented below.

Kang et al. [47] studied the relationship of plasma catecholamines, salivary alpha-amylase, heart rate, and blood pressure in psychological stress conditions. They measured plasma catecholamines, salivary alpha-amylase, heart rate, and blood pressure to monitor the stress condition of the subjects. The authors divided 33 participants into two groups: the experimental group having 16 participants that appeared in college final examinations and the control group with 17 participants. The control group did not undergo any test. Stress was measured once, anxiety levels were measured twice, and levels of plasma catecholamines, salivary alpha-amylase, heart rate, and blood pressure were measured seven times and declared a subjective stress marker. During psychological stress, the salivary alpha-amylase level changed significantly, and a partial correlation was found between salivary alpha-amylase and blood pressure, heart rate, and plasma catecholamines.

Arza et al. [49] presented a quantitative method to monitor acute stress levels of healthy people using physiological and biochemical signals. The authors used skin temperature, heart rate, EDA, ECG, and PPG as physiological parameters, while copeptin and prolactin were measured from blood samples, and cortisol and alpha-amylase were measured from saliva samples. In total, 40 volunteers participated in this experiment. The TSST was chosen for the study, to induce stress, as it provides robust and reliable acute stress induction [83]. The stress test was conducted in five different stages, i.e., storytelling, memory test, stress anticipation, video display, and arithmetic task stage. The time allocated for the whole test was approximately 25 min. The correlation analysis of physiological features showed some equal changes during the study. This study had two limitations: first, the intra-individual variability of cortisol and alpha-amylase were at the peak in morning hours; second, the alpha-amylase peaked at 10 to 15 min after the stress was induced and was assessed after 25 min by the authors. They concluded that there was no clear correlation between physiological parameters and perceived stress levels. Moreover, the alpha-amylase peak level time is 10 to 15 min after stress onset and, thus, should be measured within that time frame. The authors also mentioned the importance of the timing of biochemical collection. Alpha-amylase and cortisol were measured in the morning (at that time, intra-individual variability is the highest). Regardless of the limitations, they claimed that their study provides a solid foundation for stress monitoring as they identified a scoring function for daily stress monitoring of an individual.

Cashdan [50] explored the relationship between competitive aggression and hormones in women. The author measured total and free testosterone, cortisol, androstenedione, and estradiol during the early stages of the follicular phase of their menstrual cycle. In total, 30 women participated in this study, and all of them were university students. Each participant was called one at a time to the lab during the early follicular phase. Hormones were analyzed from blood serum. All participants competed against each other in 12 topics predefined by the authors. The participants completed dairies with eight options to score their feelings during these tests. Data on the competitors’ aggression level were recorded using their diaries, which also acted as the ground truth. Cashdan found that women with low levels of testosterone and androstenedione presented less competitive feelings, while high levels of androstenedione in women promoted more competitive feelings (mostly verbally). Moreover, estradiol levels were unrelated to any competitive feeling, although fewer competitive interactions were reported by the women having high estradiol levels.

Ahmed et al. [84] showed the association of salivary cortisol with blood pressure in patients with acute ischemic stroke. The authors measured blood pressure and pulse rate with noninvasive devices for 24 h to show how both are associated with each other. Saliva samples were collected at the time of inclusion, in the evening of the same day, in the morning of the next day, and in the evening of the next day. The total number of patients with stroke included in this study was 58. These patients were recruited after the exclusion of hemorrhage within 14 h 15 min (meantime). The authors used the Vertical Visual Analogue Scale [85] and State–Trait Anxiety Inventory by Spielberger [86] for the psychological evaluation of the patients. Each patient was asked 20 questions with four different answers. The authors concluded that there was a positive correlation between salivary cortisol and 24 h blood pressure. This also suggests that stress can be considered as a major contributing cause of high blood pressure in patients with recent stroke.

From the above review, it can be concluded that biochemical indicators of stress provide better detection and monitoring of stress, but most of these indicators can only be measured invasively. If measured noninvasively, as in some cases, the extraction of the required hormone from the sample collected takes time and, thus, cannot be used in real-time stress-monitoring devices. There is also a scope of integrating some of the biochemical indicators, such as cortisol level measurement, with physiological indicators, such as heart rate, respiratory rate, and activity monitoring, to develop a more robust and accurate device for stress monitoring. Table 4 summarizes the above-discussed papers and their conclusions.

## 4. Discussion and Conclusions

Stress arises from events that threaten the homeostatic stability of a person. The human biological system is very complex, and stress evokes different physiological and cognitive reactions in the human body [87]. For this reason, stress markers established until now do not provide any reliable assessment of the quantitative stress response. To the best of our knowledge, there is no easily applicable and repeatable method available that can compare the stress response levels of one person in different situations. Moreover, the stress response of two different persons is also different. The goal of this review paper was to find approximated quantitative measures of a person’s homeostatic imbalance.

Generally, in a stress induction experiment, questionnaires are used as a standard stress state reference. These psychometric questionnaires are not designed to be used in general applications. Moreover, these questionnaires are subject to individual variability and depend upon the person’s self-perception about their condition. Among the research community and professionals of the medical field, there is no agreement on the reference standards for monitoring stress levels and measurement methods. This lack of a standard for stress evaluation can be due to the variability in stimuli of stress, to which each human reacts differently. Furthermore, the available literature aimed to address one or few stress responses in an individual study rather than comprehensively describing the physiological stress response.

The use of biochemical markers may result in better and promising results in the detection of stress, but one of the biggest drawbacks of biochemical markers of stress levels is their relationship with the intensity of perceived stress. The reason is that this relationship between biochemical hormones and stress is both complex and understudied.

Table 3 and Table 4 summarized the types of stressors used in each study reviewed in this paper, along with the types of bio-signals collected by the authors to measure and monitor stress. It can be observed that some studies measured the same signals and used the same stressors, but the reported results showed significant differences. For example, the measured signals were the same in [39,40,43]; however, the classification accuracy achieved by each study is different (highest: 97.4%, lowest: 80%). Furthermore, the studies [35,42] also used the same signal for stress detection but reported a classification accuracy of 78.4% and 68%, respectively. In [21], the authors concluded that the physiological signals alone cannot provide acceptable accuracy for stress detection and that contextual information should also be included during the data collection. This was also evident in the results by [36], with improved accuracy of 92.4% when using contextual information. On the other hand, [56] achieved an accuracy of 81.82% by using only EDA (skin conductance), and, similarly, [37] reported an accuracy of 89.7% by using four physiological parameters and no context information.

The possible reasons for this variation in prediction accuracies can be due to the variations in the experimental setup (real and controlled environment), use of different features extracted from the raw data (time- and frequency-domain features), different lengths of recorded data (varied from 5 min to 1.5 h), different placement of sensors (chest worn, wrist-worn, and foot-worn), and the different number of subjects recruited for the experiment.

Despite the abovementioned high classification accuracies using biophysiological parameters, some studies, for example, [49] and [79], reported that there is no clear correlation between perceived stress and biophysiological parameters. Additionally, these studies suggested that biochemical markers of stress should be considered when designing a stress monitoring system. Interestingly, in [79], the authors suggested that biophysiological markers of stress (heart rate) can be a better indicator than biochemical markers (salivary cortisol, a most frequently used biochemical indicator of stress). Contrary to this, [47] and [84] suggested a positive or a partial correlation of salivary cortisol with physiological stress indicators (i.e., heart rate, heart rate variability, respiratory rate).

Reported accuracies are collected and plotted in Figure 3. It is important to note that there are different methods of feature extraction from a raw signal, as well as different ways of calculating accuracies. Commonly used tools, along with accuracy, to evaluate the performance of the different indicators and classifiers include confusion matrix, specificity, sensitivity, recall, f-score, the area under the curve, positive predictive value, negative predictive value, and likelihood ratio (positive and negative), as described in [88,89,90]. In the literature reviewed, authors might have used different matrices for stress relevant feature extraction and classification; thus, reported accuracies may not be comparable.

The variable and contradictory evidence in the literature on the use of either physiological or biochemical stress markers leads to a conclusion that neither of these biomarkers in isolation can provide sufficient means of monitoring stress. Therefore, a combination of physiological and chemical stress biomarkers, with contextual information, can be a more reliable solution for stress monitoring. A multisensor platform with data-driven personal insights can help track and intervene in cases of stress in the high-risk population. There is still a need for a novel, more sensitive, and more specific stress monitoring system that should be easily implemented and adopted by medical professionals and home-based consumers.

## Figures and Tables

**Figure 1 diagnostics-11-00556-f001:**
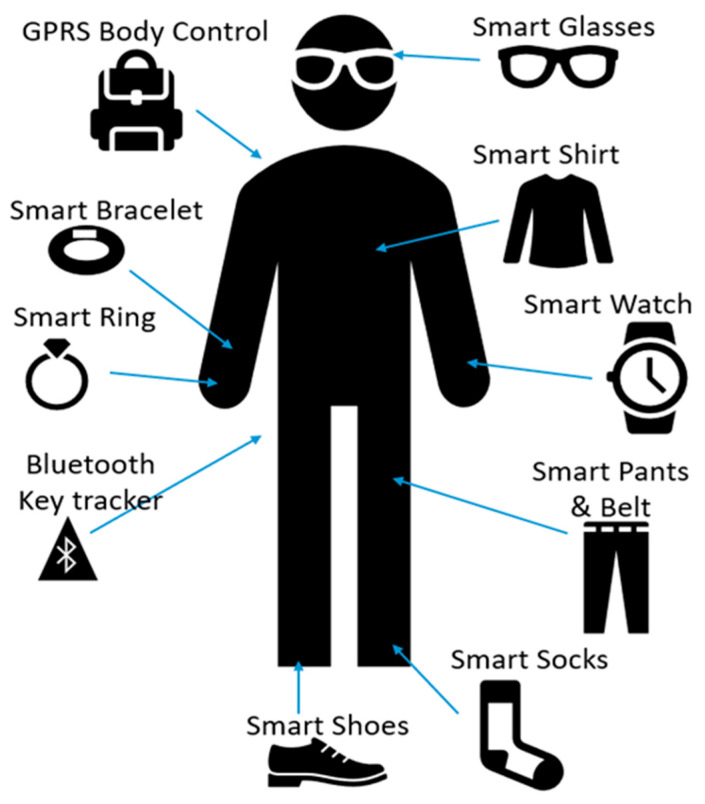
Possible sites for placement of smart sensors.

**Figure 2 diagnostics-11-00556-f002:**
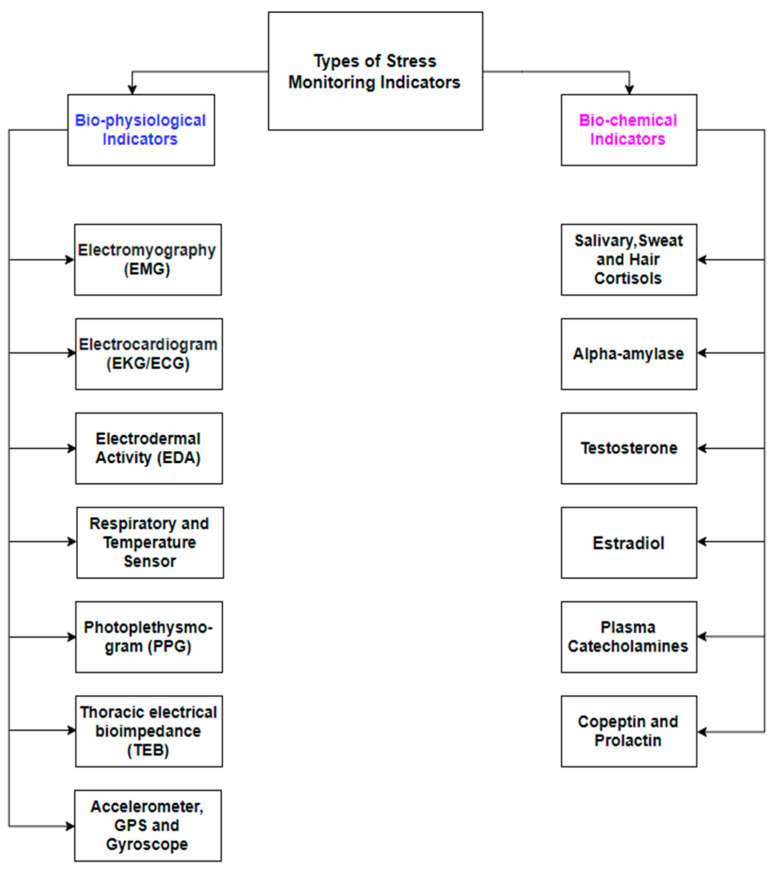
Types of stress indicators: physiological (**left column**) and biochemical (**right column**).

**Figure 3 diagnostics-11-00556-f003:**
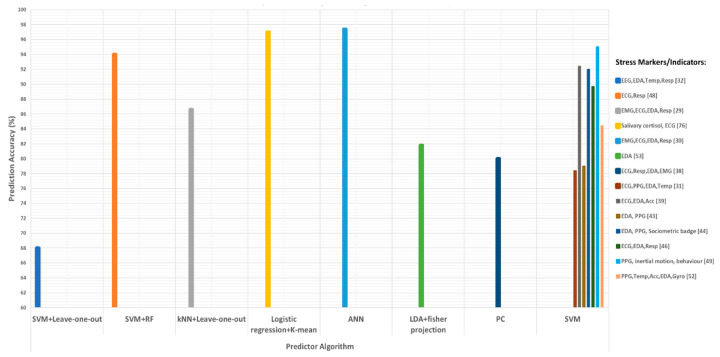
Reported prediction accuracies of various prediction algorithms using different stress indicators/markers. In the figure, SVM is support vector machine, RF is random forest, kNN is k-nearest neighbors, ANN is artificial neural network, LDA is linear discriminator analysis, and PC is principal component analysis.

**Table 1 diagnostics-11-00556-t001:** Stress assessment test and brief detail.

Test Name	Stress Assessment Method
Mental Arithmetic Test	Participants are asked to solve arithmetic questions (subtraction, multiplication) within a time frame to induce stress.
Trier Social Stress Test (TSST)	Requires participants to deliver a speech on any given topic with a short time to prepare. After the speech, the participants are also asked to perform some verbal calculation. Both tasks are performed in the presence of an evaluating audience.
Stroop Test	Participants are shown the names of different colors written in various font colors and are asked to tell the font color rather than reading the word.
Perceived Stress Scale (PSS)	Participants fill out the questionnaire by rating the questions about their feelings and thoughts. The total score varies from 0 (no stress) to 40 (highest stress).
Visual Analogue Scale for Stress (VASS)	In this test, participants are asked different questions during a given task or experiment to rate their stress on the scale as no stress, moderate stress, or high stress rather than a numerical value. Most of the time, a 5-point (smiley) scale is used for stress assessment.
Stress Response Inventory (SRI)	The Stress Response Inventory consists of 39 questions scored in the range of 0 to 156. These questions are categorized into 7 factors, i.e., tension, fatigue, depression, aggression, anger, somatization, and frustration. A high score means high perceived stress.
COPE Inventory	There are 28 items of self-reporting questions designed to measure the efficiency of the ways in which participants cope with a stressful event. A score is given to each question on a scale of 1 (low stress) to 4 (high stress). The total scoring determines the participants stress coping style, i.e., approach coping or avoidant coping.
Holmes and Rahe Stress Inventory	Measures the amount of stress incurred within the past year. Participants select events that occurred in their life from the 43 life stress-related events. Each event has different scores. Participants accumulating a score greater than 300 are at higher risk of illness, while a score lower than 150 suggests a slight risk of illness.
State–Trait Anxiety Inventory (STAI)	Participants validate 20 questions that measure the state and trait anxiety. Participants respond to the questions on a scale of 1 to 4, where 1 denotes the least stress while 4 denotes a high-stress state.
Montreal Imaging Stress Task (MIST)	MIST consists of three stages, i.e., rest, control, and experiment. In the resting stage, the participant looks at the static screen of the computer. In the control stage, the participant is asked to solve a series of mathematical problems, while, in the experiment stage, some difficult and time-constrained arithmetic tasks are given to induce high stress.
Perceived Stress Questionnaire (PSQ)	Participants fill out two types of questionnaires consisting of 30 questions; the first questionnaire has questions about stressful experiences and feelings over the last 2 years while the second one has questions about stress during the last month. Participants have to score each question from 1 (no stress) to 4 (stressed).

**Table 2 diagnostics-11-00556-t002:** Most commonly used bio-signals for stress monitoring.

S. No.	Bio-Signals	Units *
1	Skin conductance (also known as electrodermal Activity, EDA) [21,35,36,37,38]	µS
2	Electrocardiography (ECG) [19,37,39,40]	mV
3	Electroencephalograph (EEG) [41,42]	µV
4	Respiration (Resp) rate, blood pressure (BP), and blood volume pulse (BVP) using photoplethysmography (PPG) [35,43,44,45]	Breaths/min, mmHg and mV, respectively
5	Skin temperature (ST) [35,42,46]	°C
7	Electromyography (EMG) [39,40,43]	µV
8	Plasma catecholamines, copeptin and prolactin, steroids samples [47,48,49]	mcg/24-h, ng/mol, ng, respectively
9	α-amylase samples [47,49]	µL
10	Cortisol samples [50,51,52]	nmol/L

* Here, µS is microsiemens, mV is millivolts, µV is microvolts, °C is degrees centigrade, mcg is micrograms, ng is nanograms, µL is microliters, and nmol/L is nanomoles per liter.

**Table 3 diagnostics-11-00556-t003:** Biophysiological parameters (sorted year-wise).

Ref	Year	Signals	No. of Subjects	Stressors	Conclusion	Features Extracting Methodology/Classifier
[39]	2000	ECG, EMG, EDA, and respiratory	10	Garage exit, city road, toll booth, highway driving, ramp turnaround, two-lane merge, bridge crossing, and entering the garage	The combination of all features from the four type of sensors had an overall accuracy of 86.6%.	K-nearest neighbors (k-NN) classifier
[40]	2004	ECG, EMG, EDA, and respiratory	24	Some audio, visual, and cognitive stimuli	Achieved a classification accuracy of 97.4% using data intervals of 5 min and found the highest correlation between heart rate and skin conductance metrics.	Artificial neural network (ANN)
[35]	2004	ECG, PPG, EDA, and ST	50	Rest, highway drive, and city drive	Achieved a recognition rate of 78.4% for three emotional and 61.8% for four emotional states.	Support vector machine (SVM) classifier
[59]	2009	ECG, ACC, GPS, EDA, and respiratory	3	Mental arithmetic test, Stroop color–word test	The PDM algorithm shows lower inter-subject variance and interestingly showed some comparable performance between and within subjects, where PSD performance decreased when used between subjects.	Principle dynamic modes (PDM) algorithm
[42]	2010	ECG, PPG, EDA, and ST	22	Public speaking, mathematics, mental, social, and physical challenges	SVM based model detects stress with high precision and recall rate (68% accuracy), especially when they used personalized information with SVM.	Support vector machine (SVM) classifier-based model
[62]	2010	EEG, EDA, PPG, and respiratory	15	International Affective Picture System (IAPS)	Characteristics of the EEG signal were extracted by wavelet coefficients and Higuchi’s algorithm, as well as correlation dimension. An accuracy of 82.7% using the Elman classifier was achieved.	Wavelet coefficients and Higuchi’s algorithm, as well as correlation dimension with Elman classifier
[43]	2011	ECG, respiratory, EDA, and EMG	18	Perceived stress scale (PSS) questionnaire	80% accuracy indicates the suitability of used features for the detection of stress in a subject.	Principal component analysis (PCA) technique
[36]	2012	ECG, EDA, and ACC	20	Stroop color test and mental arithmetic problems based on Montreal Imaging Stress Task (MIST)	The inclusion of accelerometer data improved the stress detection process (92.4%) in a mobile environment.	Decision tree classifier, 10-fold validation, and least complex classifier
[64]	2013	ECG, EMG, GSR, and ST (only concern on ECG and HRV)	60	Stroop word–color test	Optimum features of ECG were extracted through fast Fourier transform (FFT). Accuracy of 91.66% and 94.66% was achieved using probabilistic neural network (PNN) and k-nearest neighbor (kNN) classifier, respectively.	Fourier transform (FFT) along with PNN and kNN classifier
[65]	2014	ECG, respiratory, body temp, GSR	10	Hajj pilgrimage	During sleep, the activity of the upper body and the duration of sleep contributed most to the detection of stress. The body temperature can be neglected as it did not contribute anything. Stress state was classified with an accuracy of 73% using SVM as a classifier.	Support vector machine (SVM) classifier
[44]	2015	EDA and PPG	5	Trier Scope Stress Test (TSST)	Detected stress of each participant with an average accuracy of 78.98%, i.e., combining all five participants’ stress detection accuracies using SVM.	Support vector machine (SVM) classifier
[57]	2015	ECG and TEB measurements	40	Films, game based on the addition	The measurement showed high potential in the use of ECG and skin activation (TEB) signals for the detection of long periods of stress or sudden increment in mental work overload or emotional responses of the people. With MLP classifier, authors achieved an error rate of 21.23%, 4.77%, and 32.33% for activity identification, emotional state, and mental activity, respectively.	Low-pass filtering and decimated intermediate frequency-based algorithms along with multilayer perceptron classifier
[66]	2016	EDA, PPG, and sociometric badge for recording	18	STAI (State–Trait Anxiety Inventory) and TSST	Achieved higher accuracy, i.e., 92%, with SVM (RBF kernel) classifier, as compared to linear kernel SVM (80%), AdaBoost (67%) and KNN (62%), when using selected set of features.	SVM (RBF kernel) classifier, linear kernel SVM, AdaBoost, and KNN
[37]	2017	ECG, EDA, and respiratory	14	Real driving environment	Using a full feature set, SVM with linear kernel gave the highest inter-drive classification precision. For cross-drive level, SVM with RBF kernel gave a precision score of 89.7%.	SVM with linear kernel, SVM with RBF kernel
[21]	2017	EDA, ST, ACC, and PPG	5	Randomly generated equations (solved verbally)	Without contextual information, the stress detection was not in the range of acceptable accuracy, while, when they included the context information, the detection F-score jumped to 0.9 from 0.47 and precision increased to 95% from 7%.	Time-domain features such as mean inter-beat interval (IBI) of a sample, standard deviation, square root of the mean of the squares of the differences between adjacent IBI samples, and the percentage of the differences between adjacent IBI samples that are greater than *x* ms (*x* = 20, 50, 70) along with SVM classifier
[19]	2017	ECG and respiratory	39	Montreal Imaging Stress Task (MIST)	An accuracy of 84% using random forest features and SVM classifier in discriminating three stages of stress, while, for binary classification, i.e., rest and stress, they achieved an accuracy of 94%.	Random forest features and SVM classifier
[67]	2017	PPG and inertial motion and driver behavior	28	Euro truck driving simulator version 2	Sequential feature selection with RBF kernel SVM classifier was able to achieve a classification accuracy of 95%, which shows the suitability of their glove as the driver’s stress detection device.	RBF kernel-SVM classifier
[55]	2018	ECG	1	Daily life stress	The variation of stress index shows high concordance with the work schedule of the subject and, thus, can provide an acceptable solution for comparison of stress levels of different individuals.	By combining time- and frequency-domain nonlinear features
[45]	2018	PPG, Temp, ACC, and EDA	28	City Car Driving simulator	An accuracy of 68.31% for four states (i.e., normal, stressed, drowsiness, and fatigue) and accuracy of 84.46% for three states (i.e., normal, stressed, and drowsiness or fatigue) classification.	Used pulse intervals and compared their values with standard pulse interval values. Winner-takes-all (WTA) and max-wins voting (MWV) methods were used along with SVM classifier for classification
[56]	2018	EDA only (ECG, EMG, and respiratory)	11	Driving on the highway, in the city	After using Fisher projection and linear discriminant analysis (LDA) on the data collected from the dataset, the authors claimed to achieve a classification accuracy of 81.82%.	Fisher projection and linear discriminant analysis (LDA)
[69]	2019	PPG, EDA, GSR, and ACC	21	Summer camp (training, the contest, and free day)	If individual data of each person are enough to design a person-based model, it should be developed; otherwise, each person should be clustered in accordance to their behavior in stress and then a clustering model should be developed to increase the classification accuracy of the general model.	Clustering model like kNN
[54]	2019	EDA	1	Driving on the highway, in the city	From the experiment, the authors concluded that their classification results indicate that road type and traffic conditions are important features related to driving stress. The authors reported an accuracy of 80.3% with a sensitivity of 85%, a specificity of 78%, and positive predictivity of 70% while using logistic regression as a classifier.	EDA from Empatica E4 watch and logistic regression-based classifier
[38]	2020	Galvanic skin response (GSR)	10	Predefined PYSIONET dataset and driving on the highway, in the city	For the classification, the authors designed a binary logistic regression model and achieved an overall accuracy of 85.3% on data from PYSIONET, while they achieved 83.2% accuracy on the validation data, analyzed through cross-validation. The authors also proposed that their developed model can be embedded in existing wearable GSR sensor devices and, thus, can enable detecting and monitoring the driving stress in real time.	GSR from Empatica E4 watch and binary logistic regression-based classifier along with cross-validation technique

**Table 4 diagnostics-11-00556-t004:** Biochemical parameters (sorted year-wise).

Ref	Year	Signals	No. of Subjects	Stressors	Conclusion
[50]	2003	Total testosterone, free testosterone, estradiol, androstenedione, and cortisol	30	Early follicular phase	Women with low levels of testosterone and androstenedione presented less competitive feelings. Moreover, estradiol levels were unrelated to any competitive feeling.
[51]	2004	Salivary cortisol (in the blood)	12	The psychosocial stress test, Trier social test (TSST), and reading test	The level of salivary alpha-amylase was significantly lower in smoking females than non-smokers, while it was higher in smoking males than in non-smokers. Identified that the production of salivary cortisol affected the association of norepinephrine and amylases. Activation of parasympathetic nervous systems decreased the overall saliva production and volume. Therefore, the volume of saliva and amylase levels should be measured relative to the saliva produced.
[84]	2004	BP, pulse rate (PPG), and saliva	58	Vertical Visual Analogue Scale (V-VAS) and State–Trait Anxiety Inventory	There was a positive correlation between salivary cortisol and 24 h blood pressure.
[52]	2008	Cortisol level and EMG	16	Arithmetic problems, tasteless gum	Fast chewing had a greater effect on the stress release than a slow chewing rate, while the integration of EMG signals did not show any major difference in the 3 chewing rates.
[47]	2010	Salivary alpha-amylase, plasma catecholamines, BP, and HR	33	College academic final exam	The salivary alpha-amylase level changed significantly, but a partial correlation was found, statistically, between salivary alpha-amylase and blood pressure, heart rate, and plasma catecholamines.
[73]	2012	Hair cortisol	-	Daily life stress (3 months)	Identified some gaps in the currently available literature: firstly, to clarify the mechanism underlying cortisol incorporation into hair, and, secondly, to determine the factors that cause variation in hair cortisol such as the effect of hair washing.
[78]	2013	Sweat and saliva samples	17	Intense exercise	Intense exercise could increase the concentration of cortisol in hair, which was not decreased by hair washing.
[48]	2016	Steroid hormones in hair	40	Perceived Stress Questionnaire (PSQ)	The concentrations of steroids in the hair were a decisive predictor of the increased long-term HPA axis. Furthermore, this biomarker could capture stress even after burdening events or any physical activity was finished.
[81]	2018	Biochemical (salivary cortisol) and physiological (HRV measures) domains	30	Academic final examination, Psychological Stress Response Inventory	The salivary cortisol levels were negatively correlated with the HRV indicator of parasympathetic activity, while they were positively related with the HRV indicator of sympathetic activity. The results also showed that the value of the mental stress index (MSI) was very sensitive to acute stress and could predict stress with an accuracy of 97%.
[49]	2019	ST, HR, pulse wave (EDA, ECG, PPG), copeptin, prolactin (blood), cortisol, and alpha-amylase (saliva)	40	Trier Social Stress Test (TSST)	There was no clear correlation between physiological parameters and perceived stress levels. Moreover, alpha-amylase peak level time is 10 to 15 min after stress onset and, thus, should be measured within that time frame. Alpha-amylase and cortisol were measured in the morning (at that time, intra-individual variability is high).
[79]	2019	PPG and endocrine (salivary) cortisol	32	Childhood Trauma Questionnaire (CTQ)	No significant effect of early life stress on heart rate (autonomic indicator) and salivary cortisol (endocrine indicator), but the authors suggested that heart rate is a better indicator (of stress) than salivary cortisol as it is more sensitive to individual stress reactivity than salivary cortisol.
[76]	2019	Oxy-hemoglobin (oxy-Hb)	4	Decision-making and memory recall tests	Whenever high stress occurred, the average difference value of oxy-hemoglobin (oxy-Hb) increased.

## Data Availability

Not applicable.

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
