# Peer review of "A Review of Biophysiological and Biochemical Indicators of Stress for Connected and Preventive Healthcare"

_diagnostics, 2021, doi:10.3390/diagnostics11030556_

Round 1
Reviewer 1 Report
The article is a fairly detailed review of biochemical and biophysical methods for assessing stress. I have a few comments on the structure of the article. It seems to me that Tables 2 and 3 should be located in the appropriate sections on biophysical and biochemical methods for monitoring stress, and not in the discussion of the results. The same applies to Figure 3. References to literature sources are numbered in the order of their mentioning in the text, apparently the authors did not manage to make changes. Do biophysical and biochemical methods of stress monitoring combine? If so, it would be appropriate to write about it. If not, why not.
Reviewer 2 Report
Biosensors have drawn increasing interest for their potential as diagnostic tools for human healthcare. It was with interest then that I read this timely manuscript submission focused on their use in the area of stress detection. What follows is in the spirit of turning a fair review into an excellent review.
Overall, the writing quality is good. The manuscript would benefit from a thorough editing by one well-versed in written English. Specifically, the use, or lack thereof, of well-placed and appropriate prepositions is encountered throughout. There are also multiple cases where the singular vs. plural form of nouns is incorrect. Neither of these issues is insurmountable, as MDPI offers excellent editorial assistance as part of the publication process.
Specific Suggestions. The paper suffers from too many details that are not adequately explained for non-engineers. At the risk of being perceived as preaching: a review should be more than just a recitation of the details of one paper after the next on a topic. For starters, one wonders who the intended audience is for this offering? Is it for healthcare workers? Engineers? Bioengineers? Statisticians? Basic scientists? Experts on biosensors? (It does not seem to be written for this basic scientist who studied stress mechanisms in the laboratory for many years- and who freely admits to perhaps not being the most appropriate person for this review assignment). Generally, one wants to target a Review to the broadest possible audience of busy science-literate professionals who gravitate toward reading reviews so they can get “up to speed” on a topic in a short amount of time. (Experts on a particular topic have already read the primary literature in their field.) Thus, those who author Reviews must have the souls of teachers and fill in a range of knowledge gaps for a range of readers. Enough said, except maybe the authors might want to consider bringing a biologist on board as a co-author.
Assuming the authors are interested in reaching a broader audience, the addition of four paragraphs to the Introduction is humbly recommended.
- A detailed discussion of the biophysiological and biochemical stress indices included in Table 1 and how they are measured. Without going into too much detail, it should be expanded to at least include the specific types of data obtained with each, how each of these relates to the physiological condition called “stress” (e.g. what is alpha-amylase) and what specific unit of measure describes each one. (i.e., when the data are entered into the spreadsheet how are they labelled: pg/ul? mV? l? mmHg?).
- A slightly more detailed description of the psychological measures of stress (e.g., Stroop test mental arithmetic) that are used to characterize the subject populations would be a good addition. Perhaps another Table would work here.
- A much-expanded discussion of the “machine learning algorithms” that are alluded to from lines 63-67 is in order. This aspect of the entire paper really needs some elaboration if you intend it to be read by non-engineers. How are data, obtained through any of the methods described in part one, transmitted (or transcribed?) to an artificial neural network or a Baysian network that uses machine learning or other computer aided diagnostic tools? Are the subjects physically connected to computers? Do the sensors transmit via Bluetooth? What exactly are these and why are they useful for analyzing the data gained from devices described in part 1? Is some of this regression-capable statistical software? Along these lines, the paper is full of references to technical approaches never explained for non-engineers, e.g., SVM, probabilistic neural network, k-nearest neighbor classifier.
- Finally, a discussion of the statistical (or other) approaches used to analyze these data is in order. This field describes outcomes in terms of “percent classification accuracy.” How is Accuracy-usually a measure of how close a value is to the true value- computed or determined for work such as this? Is this related to parametric statistics in any way? How does one interpret accuracy values? Is there a minimum acceptable accuracy?
Other observations. The paper needs to be scanned to determine that all acronyms are defined where they are first encountered in the manuscript, e.g., support vector machine, line 66. They do not need to be defined, and should be referred to as fully spelled out, if they are never used again in the paper.
What is “ground truth”? Please define.
The paragraphs inserted between lines 139-163, unfortunately, feature reference numbers (79, 80) that are out of order. EndNote or some other citation manager software is a worthwhile investment for coping with reference issues such as this.
There seem to be many papers targeted toward understanding the stress responses in drivers of vehicles (as opposed to software drivers). Why is this? One imagines vehicular accident statistics play a role? One would think driving tasks in such studies should include parallel parking during the driver’s exam for a peak stress response…
The amount of literature covered is expansive. Clearly the authors have done a lot of work to put this together.
Round 2
Reviewer 2 Report
Second Review of Iqbal et al.,
This revised manuscript is enormously improved over the original version. The authors are to be congratulated on a review that now largely bridges the gap between the academic disciplines of engineering vs. those of biology/psychology with these revisions. Many will find this a useful contribution to the literature.
Specific Improvements: The addition of Tables 1 and 1a are especially welcome, given the large number of acronyms in the paper. The expansion of the paragraph between lines 60 and 77 describing how the sensors communicate with computers for the subsequent machine learning-based analyses is excellent. The definition of “ground truth” is also very helpful as it appears to be fundamental to this field. The paragraph from lines 88 through 107 describing the relevance of the different biological measures and the addition of Table 2 listing the units of measure for the various sensors should make this review a popular reference for those interested in this topic.
For those new to the world of machine learning, the discussion of Accuracy in Appendix B is very useful information as are lines 651-686 in the Discussion. One still wishes this information was placed further forward in the review. However, this can also be easily addressed by inserting a phrase at the first mention of this concept earlier in the paper directing the interested reader to the Discussion and Appendix B. (see Discussion and Appendix B for more information concerning the concept of “Accuracy”). An additional citation here that specifically describes how machine learning classification is evaluated might also be a wise. Conversely, there are also some discussions on the web that appear useful: e.g. https://www.jeremyjordan.me/evaluating-a-machine-learning-model/.
The paper as a whole still needs a thorough editing by someone well-versed with written English to eliminate numerous grammatical errors.
Unfortunately, three serious errors appear: Lines 102-103 refer to copeptin and prolactin as a “by-products” (not a biological term, one would use “metabolite” or “molecular derivative” perhaps) of AVP. This is incorrect. Copeptin is a component of the pre-pro-vasopressin precursor peptide, as is AVP. Prolactin is unrelated to AVP and copeptin in terms of cellular origins, molecular structure, function, and synthetic pathways.
Line 472 incorrectly equates salivary cortisol with alpha-amylase. Although they are both stress indicators and both are found in saliva, they are derived from different sources in the body. See Cozma et al., 2017, Braz J Med Res 50(2) for a good overview.
Line 540 incorrectly states that Bonke et al., used commercially available fMRI in their study of early life stress. fMRIs cost hundreds of thousand of dollars and are only affordable by hospital radiology departments.
Line 567. This refers to single sensor studies despite being in a “multiple sensor” (line 566) subdivision of the paper. Double-check for correctness.
Figure 3., which is a great summary, appears a bit truncated on the bottom.
Lines 651-686. This is a nice discussion of accuracy. Is there a cut-off below which one would view the results as insignificant or inconsequential?
Author Response
Please see the attachment.

This manuscript is a resubmission of an earlier submission. The following is a list of the peer review reports and author responses from that submission.
Round 1
Reviewer 1 Report
This paper presents a review of various physiological and chemical indicators of stress, as reported by various authors. Overall, this is a very important topic and the authors have identified numerous important literature for this review.
My main concern with the review paper is threefold:
First, the authors provide a summary of various literature, but they do not provide a clear understanding of the differences in the research of when similar types of sensors have been used by different researchers. Ideally, rather than individually reporting about each paper in section 3, the authors could have grouped the papers based on some similar topic idea -- e.g., they could have lumped all papers that used ECG together and show how each paper differs from others.
Second, I am concerned about how the 62 papers were identified. in my opinion, given the large history of literature involving stress detection, using only 62 papers is an extermenly small sample size. However, even for these 62 papers, and the variety of keywords, I am curious to know the number of papers each keyword or search term provided. Did the authors choose the top-20 papers for each keyword? Without such information, it is challenging to know if the search was comprehensive.
Third, the paper looks incomplete in various aspects -- e.g., in table 1, only some of the bio-signals have citations. Ideally, each bio-signal should have its citation (even if it is a repeat citation of another bio-signal's citation). Another question that arises in section 3 is where did the authors obtain the ground truth from? A third question that arises is how do the authors compare across a plethora of evaluation metric used by the original papers.
Given the severity of these limitations, I believe that the authors will have to significantly modify their paper and thus I believe it is not ready for publication at the moment.
Reviewer 2 Report
In Section 3.1, the authors just arranged the papers' abstract without any categorization or review points. The authors already provided the search methodology in Section 2. That needs to be extended in Section 3, not just show papers' abstracts.
Authors missed many regarding papers. Bellows are some examples:
1. Identifying Traffic Context Using Driving Stress: A Longitudinal Preliminary Case Study HKK OV Bitkina, J Kim, J Park, J Park Sensors 19 (9), 2152
2. Development of a statistical model to classify driving stress levels using galvanic skin responses J Kim, J Park, J Park Human Factors and Ergonomics in Manufacturing & Service Industries
In Table 2, readers would interest the processing methods.